# Predicting Intention to Participate in Community Physical Activities for Adults with Physical Disabilities

**DOI:** 10.3390/jpm12111832

**Published:** 2022-11-03

**Authors:** Qi Xu, Hongwu Xie, Dingzhao Zheng, Xinhong Wu, Yun Zhang, Taibiao Li, Tiebin Yan

**Affiliations:** 1Department of Clinical Medicine, Xiamen Medical College, Xiamen 361023, China; 2Department of Rehabilitation Medicine, Fifth Hospital of Xiamen, Xiamen 361101, China; 3Department of Rehabilitation Medicine, Sun Yat-sen Memorial Hospital, Sun Yat-sen University, Guangzhou 510120, China

**Keywords:** international classification of functioning disability and health, short form 36 health survey questionnaire, participation intentions, community physical activity, adults with a physical disability, theory of planned behavior

## Abstract

Structural equation modeling was used to derive a relationship predicting the intention to participate in community physical activity among community-dwelling adults with a physical disability in Xiamen, China. The data were collected in a cross-sectional survey. The structural equation modeling combined biomedicine and the theory of planned behavior. It integrated ratings using the rehabilitation set from the international classification of functioning, disability, and health and role-physical scores from the short form 36 health survey questionnaire instrument. The model demonstrated a good ability to predict self-reported participation intentions, explaining 62% of the variance. The standard coefficients showed that activity limitation (27%), role-physical score (21%) and body impairment (14%) were the most influential predictors. ICF-RS ratings and role-physical ratings together can usefully predict physically disabled adults’ intention of participating in community physical activities. Suggestions are presented for multidisciplinary intervention and improving this portion of the WHO’s classification system.

## 1. Introduction

Participation in community physical activity benefits the participants’ health. In this study the term community physical activity refers to the mass physical activities like group dancing in public places popular in China. Its community aspect encourages regular attendance and more activity than they might do on their own. Individuals with a physical disability are less likely to participate in community physical activity, and when they do they face numerous physical and social barriers [1]. That makes it important to understand the factors that influence their intention to participate.

Any such exploration of intentions should start with the “disability paradox”: two individuals with structural and functional impairment of the same severity may demonstrate different levels of disability. The underlying reason for the paradox is that behavior as well as pathology determines disability [2]. After a stroke, for example, individuals with a stronger sense of self-reliance and control over their bodies report fewer post-stroke symptoms, and they also focus more on their physical rehabilitation and lifestyle changes [3]. Biomedicine regards disability as a result of pathology and employs impairment-based biomedical models [4], while psychology interprets disability as a result of behavior and applies models of motivation such as social cognition theory [5] or the theory of planned behavior [6,7]. The primary objective of this study was to understand the relationship between impairment pathology and behavior among adults with physical disabilities by integrating biomedical and planned behavior models.

The tool used for evaluating impairment and its influence was the WHO’s international classification of functioning, disability, and health (ICF). It attempts to synthesize biomedical and psycho-social approaches to understand their mediating roles in disability. In the ICF approach disability is “…an umbrella term for impairments, activity limitations, and participation restrictions” [8]. One subset of the ICF is its rehabilitation core set (ICF-RS) which is widely used to evaluate functioning and disability in clinical rehabilitation settings [9,10]. A Chinese version of the ICF-RS has been developed which makes the ICF-RS a standardized international language for the exchange of health and function information between China and the rest of the world [11,12]. The psychometric properties of that Chinese version have shown good reliability and validity in multi-center studies, and it is now widely accepted as a functional assessment tool in China [13,14]. This study used the ICF-RS in Chinese.

However, the ICF is designed to emphasize the pathology of health and impairment. It pays little attention to the motivations underlying behavioral intentions. So, the more concrete objective of this study was to study the ICF-RS together with the motivational factors of a behavior model. The theory of planned behavior is the most popular and influential model for predicting human social behavior or behavioral intentions [6]. According to the theory of planned behavior, behavior can be predicted from intention and self-perceived behavior control ability. Although intention and behavior are usually strongly correlated, they can diverge considerably. The theory of planned behavior thus focuses on predicting intentions, which can be predicted from self-perceived behavior control, attitudes, and subjective norms [6,15]. This study applied the theory of planned behavior to focus on intention and self-perceived behavior control as motivators. The aim was to predict the intention to participate in community physical activity and study its relationship with behavior control and ICF-RS classifications.

In applying the theory of planned behavior, behavioral intentions can be assessed by simply asking people whether they will attempt to engage in a behavior, expect to engage in it, and so on [6]. In this study, the behavior in question was community physical activity, so the participants were asked whether they were willing to engage in community physical activity and whether they expected to have an opportunity to do so. Perceived behavior control here was the self-perceived ease or difficulty of successfully engaging in community physical activity [16]. This study used the role limitations caused by physical problems (role-physical) domain of the short form 36-item health survey questionnaire (SF-36) to represent perceived behavior control. The role-physical domain evaluates four types of self-perceived physical impacts: limited task endurance, accomplishing less, limited types of tasks, and task performance difficulty.

Research has shown that the theory of planned behavior can effectively link motivational and cognitive factors in explaining intentions, but that work was mostly with general populations such as seniors or persons with a chronic disease. Populations with various physical disabilities have received less attention [17,18]. Sur, Jung, and Shapiro meta-analyzed the research conducted between 1986 and 2021 and found only eight studies of adults with physical disabilities. Four of them were specifically limited to persons with a spinal cord injury; only three studies dealt with various physical disabilities. Some studies have observed physical activity as a terminal factor [7] but none which investigated physical limitations as a factor influencing intentions has been published. This study has therefore tested the idea that physical activity limitation might influence disabled persons’ behavioral intentions. An interesting finding from that review was that intentions toward physical activity among adults with a disability appear to be influenced more by perceived behavior control than by attitude or social norms. This is different from people without a disability whose intentions are mainly shaped by attitude. The difference could arise because physical limitations and related obstacles prevent adults with a physical disability from participating successfully, leading them to develop their intentions by a mechanism different from that of people without a disability [19,20,21]. This study, therefore, focused on self-perceived behavior control.

A group led by Johnston first started integrating biomedical ICF and behavioral predictors to conceptualize disability [22]. Later, integrating the ICF with behavioral models of disability was found to better explain and account for more of the observed variance than either of those predictors alone (at least with orthopedic patients) [15,23]. However, those studies were limited in evaluating only a small set of bodily impairment factors such as pain and joint stiffness. It was suggested that considering multiple factors and possibly using the ICF core sets for specific diagnoses would give better results [2,24]. This then has been the first published study to evaluate multiple body impairments and activities limitations using the ICF’s rehabilitation core set in building an integrated biomedical and behavioral model.

The study’s rationale is summarized in Figure 1.

Previous research has demonstrated interactions among ICF components: body function (**also** referred as body impairment, or structural impairment) can affect activity limitation, which in turn are predicts the functional ability of the people with diabetes [25], or the risk of falls at home [26]. One hypothesis tested was, therefore, that the potential factors of the ICF-RS might affect each other (Hypothesis 1). The data confirmed the general belief that impairments restrict participation [1]. The study therefore also tested the idea that ICF-RS factors could affect the intention to participate in community physical activities (Hypothesis 2). According to the theory of planned behavior, self-perceived behavior control can predict intentions [6,15], and previous research has demonstrated that behavior control perceptions predict the specific intention to exercise [27], or the intention to be physically active among older adults [28]. This study also therefore hypothesized that role-physical (perceived behavior control) ratings could predict participation intentions (Hypothesis 3). As a domain of the short form 36 instrument, role-physical means the role limitations caused by physical problems, in terms of ICF-RS factors affecting role-physical scores that have not previously been investigated, so this study did so (Hypothesis 4). Figure 2 presents a hypothesized path model for predicting intention to participate in community physical activities.

## 2. Materials and Methods

### 2.1. Participants

This study was a cross-sectional survey conducted between May and December 2019 in the Xiang’an district of Xiamen city in southern China. The survey and assessment were administered face-to-face. Before collecting the data, 27 medical staff of Xiamen No. 5 Hospital received professional training in ICF-RS measurement and interviewing skills. They were then organized into groups which surveyed each village and town in the district in batches. They first contacted the person in charge of the local health center, and through their introduction established a cooperative relationship with any local physicians. They explained the purpose of the investigation and the inclusion criteria to the local physicians, who then contacted the local people with a disability or/and their families, explaining the purpose of the investigation and obtaining their consent and cooperation. They then made an appointment for the interview at the local health center. The trained hospital staff conducted the interviews.

All people with a disability in each village and town of Xiang’an district were initially considered for investigation. They were screened using the inclusion and exclusion criteria shown in Figure 3. That resulted in a total of 516 adults with physical disabilities being included. Their demographics are described in Table 1.

### 2.2. Instrument Scaling

The three ICF-RS domains used (detailed in the Appendix A Table A1) were body impairment (9 items), activity limitation (14 items), and participation limitation (7 items). Each item was scored on a scale of 1–4 where 1 indicated normal functioning, 2 mild dysfunction, 3 moderate dysfunction and 4 severe dysfunction. In addition, 8 indicated not specified, and 9 not applicable. The response options 8 and 9 were considered as missing data since they do not belong to the ordinal scale from 0 to 4. Missing values were replaced with the item’s median value [29]. Role limitations caused by physical problems (role-physical) is an item domain from the Chinese version of the American 36-item Short Form Health Survey [30,31]. The role-physical domain assesses task endurance, ability to accomplish, limits on types of tasks, and task performance difficulty as physical impacts of disability. Each role-physical item was scored as 1 (not affected) or 2 (yes, affected).

As for intention to participate in community physical activity, the interviewers simply asked each subject whether they were willing to engage in community physical activity and whether they expected to have a chance to do so. Those replies too were scored as 1 (yes) or 2 (no).

### 2.3. Statistical Analyses

Version 22 of the SPSS software suite (SPSS Inc., Chicago), and version 1.1.0 of the Scientific Platform Serving for Statistics Professionals (Suzhou Zhongyan Network Technology, China) were used in the data analyses [32].

The first step was exploratory factor analysis, which verified the number of potential variables and their relationships by applying varimax rotation [33]. A structural equation modeling graphical description was then constructed on the basis of the ICF and the theory of planned behavior. Model fit was then evaluated using chi-squared goodness-of-fit (χ^2^/degrees of freedom), a goodness-of-fit index (GFI), the root mean square approximation error (RMSEA), a comparative fit index (CFI), a normed fit index (NFI), and an un-normed fit index (NNFI). The maximum likelihood model was used for parameter estimation [34].

## 3. Results

### 3.1. Exploratory Factor Analysis

Exploratory factor analysis with varimax rotation was used to identify the principal components of the observed variation. The Kaiser-Meyer-Olkin test statistic (KMO = 0.917 (>0.7), *p* ≤ 0.001) and Bartlett’s test indicated that the data set was adequate and appropriate for use in the analysis. Five components or factors which fitted well were identified. Although the ICF-RS has three official domains (impairment, activity, and social participation), b134 sleep functions, b152 emotional functions, and b280 sensation of pain were found to be strongly loaded together as one factor. That accords with the observations in clinical practice where pain may cause psychological distress and sleep problems [35]; psychological and physiological components of the emotional experience may mediate in the path from impaired sleep to greater pain intensity [36], and sleep is a physio-psychological phenomenon [37]. Those three factors were therefore grouped as a latent factor named physio-psychological reaction in this study. A loading score of 0.50 or higher was assigned to four independent latent factors: body impairment, activity limitation, physio-psychological reaction, and role-physical score.

### 3.2. Structural Equation Modeling

The potential variables and factors from the exploratory factor analysis were added as paths explaining the hypotheses (Figure 4): (H1-1) body impairment would affect activity limitation; (H1-2) activity limitation would affect physio-psychological reaction; and (H1-3) body impairment would affect physio-psychology. Bodily impairment would affect intention to participate in community physical activity (H2-1), as would activity limitation (H2-2) and physio-psychological reaction (H2-3), and role-physical score would predict it (H3). Body impairment would affect one’s role-physical score (H4-1), as would activity limitation (H4-2) and physio-psychology (H4-3). All of these relationships could demonstrate positive correlation, since in each case a lower score indicates better functioning or a better condition.

Good model fit required respecifying the hypothesized model. The directions of the relationships among the variables were maintained, but non-contributing variables were deleted. The result was a five-factor model consisting of 18 items (Figure 4), and it showed good model fit: χ^2^/df = 2.73, GFI = 0.932, RMSEA = 0.058, CFI = 0.955, NFI = 0.932, NNFI = 0.945. Intention to participate in community physical activities was the dependent factor. The standardized item loadings for the variables ranged from 0.45 to 0.89, indicating that items from both structures contributed heavily to the constructs being measured (Table 2).

Figure 4 shows that the following significant relationships support hypotheses. H1: body impairment affects activity limitation (β = 0.52, *p* ≤ 0.01); body impairment (β = 0.19, *p* ≤ 0.01) and activity limitation (β = 0.19, *p* ≤ 0.01) both affect the physio-psychological reaction. H2 and H3: 48% of the variance in intention to participate was directly explained by activity limitation (β = 0.27, *p* ≤ 0.01) and the role-physical scores (β = 0.21, *p* ≤ 0.05); H4: 22% of the variance in the role-physical scores was directly explained by physio-psychological reaction (β = 0.22, *p* ≤ 0.01).

However, bodily impairment and physio-psychological reaction were not significant direct predictors of intention to participate in community physical activity. Moreover, neither bodily impairment nor activity limitation significantly affected the role-physical ratings.

The total result of the integrated model predicting intention to participate is the sum of direct and indirect relationships. Body impairment predicts activity limitation (0.52), and then carries through (0.27) to predict intentions. This is the indirect effect of body impairment on intentions (0.52 × 0.27 = 0.14). Activity limitation (0.27) and role-physical scores (0.21) are both useful direct predictors of intention. If the small indirect relationship between ICF factors and role-physical scores is neglected, ICF-RS ratings and role-physical scores together strongly predict intentions (0.14 + 0.27 + 0.21 = 0.62). This means the integrated model can explain 62% of the variance in intentions to participate in community physical activity.

## 4. Discussion

### 4.1. The ICF-RS and Planned Behavior Together Can Predict Participation Intentions

The data show that ICF-RS ratings and ratings of role limitations caused by physical problems can together usefully predict disabled adults’ intentions to participate in community physical activity. Indeed, each alone has significant predictive power. Impaired bodily functioning, however, is not a significant direct predictor. Role-physical scores act as mediators in the joint ICF-RS and theory of planned behavior model.

Most prior research has predicted such intentions based on a theory of behavior or social psychology. A systematic literature review reported that 89 intervention studies improved the motivation for physical activity using behavior change techniques and modes of delivery theory [38]. A few scholars have integrated simple ICF framing with a theory of perceived behavior to predict activity limitations, but they considered only impairment factor of the ICF as potential predictors. They found that the behavior factors were significant predictors of activity intention, but impairment was not [2,15,23]. This study’s data are consistent with those findings. Bodily impairment does not affect intentions directly, but it indirectly impact activity intentions through other factors of the ICF such as activity limitation and physio-psychological reaction.

Reedman’s group has used ICF impairment, limitation, and restriction factors with domains from the theoretical domains framework to identify physiotherapist-perceived barriers to physical activity participation among children with cerebral palsy and developed a specific intervention plan with the behavior change wheel [39]. Although that study did not explore the relationship between the ICF and behavior, it inspires the development of specific intervention schemes based on the ICF and theories of behavior in the future.

The World Health Organization acknowledges that the ICF needs to be modernized and it encourages anyone to submit evidence-based proposals for doing so. One main concern is that many people still take the ICF as a purely medical representation, instead of bio-psychosocial [40]. Dekker has summarized models incorporating categories of psychological adjustment into the ICF and found that emotional, cognitive, and behavioral responses are major psychological categories that can usefully be integrated [41]. This study’s findings demonstrate that integrating psychological adjustment into the ICF model is important.

Although the intention to participate is a behavioral intention, it is important to include ICF-RS items in predicting it, since activity limitation directly (27%) and body impairment indirectly (14%), together account for 41% of the variance in physical activity intentions. It is also essential to consider the theory of planned behavior because the role-physical scores, representing self-perceived control of behavior here, account for 21% of the variance in physical activity intentions. The data suggest that the ICF-RS framework should be integrated with a behavior model to best predict physical activity intentions. Neither should be neglected in clinical rehabilitation assessment and practice with adults with physical disabilities, considering the ultimate goal of rehabilitation is a return to activity in the community.

### 4.2. Cognition May Mediate between Impairment and Behavior

Interestingly, role-physical was found to be directly related to the physio-psychological reactions of the ICF-RS and indirectly to activity limitation and body impairments. At the same time, role-physical scores directly predict the intention to participate. Thus role-physical results mediate between the ICF-RS scoring (biomedical and based on body impairment) and the theory of planned behavior. As role-physical scores are impressions reflecting the limitations caused by physical problems, this supports other research findings that cognition may mediate between impairment and behavior [42]. This study’s data have further clarified that cognition may mediate between ICF-RS ratings of impairment, activity limitation, and physio-psychological reactions and behavioral intentions.

### 4.3. Body Impairment May Not Directly Affect Behavioral Intentions or Behavior

It is easy to imagine how a diagnosis of bodily impairment could play a major role in influencing behavioral intentions and role-physical results, but this study’s data show that bodily impairment does not always affect participation intentions directly. Nor does it necessarily affect role-physical scores directly. However, activity limitation and any physio-psychological reaction to it do directly affect participation intentions, at least for community physical activity. This could explain the “disability paradox” to some extent. For a given degree of physical impairment, if an activity is less limited an individual has a better role-physical rating, which may allow for a stronger intention to participate in community physical activities. Such intentions may reduce the degree of disability exhibited in social situations. The data, therefore, suggest that rehabilitation focused entirely on bodily impairment risks missing other important factors compromising its effectiveness in terms of returning the disabled to the community. Physical therapy may improve ICF-RS evaluations while role-physical results can be better improved by physio-psychological interventions according to this study’s model. That may require multidisciplinary cooperation with help from psychotherapists or/and social workers.

### 4.4. Potential Improvements to Aspects of the ICF

Studies of ICF applications have been proliferating [43], leading many critics to say, for instance, that the ICF still relies excessively on a biomedical approach without sufficiently emphasizing the psychosocial aspects of disability [44,45]. The findings here support that criticism: the ICF-RS covers physio-psychological reactions including sleep, emotion, and pain, but together those items did not contribute significantly to the variance in participation intentions, while role-physical scores, a psychological factor not belonging to the ICF-RS were a significant predictor. This suggests that integrating more psychosocial factors could improve the ICF’s treatment of the psychological and social aspects of disability.

Another criticism of the ICF is that the personal factors do not properly target functioning and health, and scholars have suggested an alternative ICF scheme with personal factors more directly related to activity limitation, body impairment and participation. Indeed, participation should have a central role [45]. The findings of this study suggest that role-physical scores could be developed as personal factors because they relate closely to limitations, impairment, and participation. By specifically targeting participation in community physical activity, these findings may inspire the development of a new ICF scheme better emphasizing the mediators and personal factors.

Another significant deficit of the ICF is that it is not implemented easily [46]. When this study applied the ICF-RS targeting, specifically the intention to participate in community physical activity, only some parts of some ICF-RS items were involved in the explanation. This supports Jette’s suggestion [46] that the ICF should include more dynamic factors that influence movement in different states of functioning and disability. The ICF classifications could be usefully supplemented by designing a dynamic model of factors based on statistical analysis like the exploratory factor analysis used in this study.

### 4.5. Clinical Implication of the Study

The study’s model integrating a psychological component usefully predicts disabled adults’ intention to participate in group physical activities. The model makes it clear that activity limitation and self-perceptions of ability are more influential than actual impairment in motivating such intentions. This explains the “disability paradox” to some extent and inspires more attention among clinicians to psychological and social factors and more multidisciplinary cooperation in encouraging community participation among the disabled. At the same time, the results encourage the need for improving the international classification and supplementing it when necessary.

### 4.6. Limitation of the Study

It is important to note that only about a quarter of the disabled people contacted were eventually included in the analyses because many could not conveniently come to the local health center for assessment and interview. The sample may therefore only reflect the physically disabled population in China who can conveniently visit a local clinic. It is also important to bear in mind that the associations observed among ICF-RS ratings, role-physical ratings, and intention of participation were correlated, not causal. They need to be further validated using longitudinal data. Moreover, although the interviewers and data collectors were all specially trained, bias could still have intruded, since no one was blinded to the information gathered from the ICF-RS assessment and much was a self-reported recall. Besides, this study did not evaluate attitudes and subjective norms, which are part of the theory of planned behavior. Future studies should consider measuring this aspect of the theory. Finally, this study was part of a larger community sample seeking to understand more about the health status of persons in rural Xiamen. Only part of it is discussed here.

## 5. Conclusions

Disabled adults’ intentions to participate in community activities should be given closer attention in their rehabilitation. An integrated ICF-RS (biomedical) and role-physical (behavior) model can be useful for predicting such intentions. The data suggest that activity limitation, role-physical score, and body impairment are the factors contributing most to forming intentions, so they should be the target of interventions. Rehabilitation based on the ICF-RS and multidisciplinary intervention is suggested. The good fit of the integrated model should inspire clinicians to emphasize psycho-social factors along with biological ones and to develop a new ICF scheme or a supplement to the ICF classification.

## Figures and Tables

**Figure 1 jpm-12-01832-f001:**
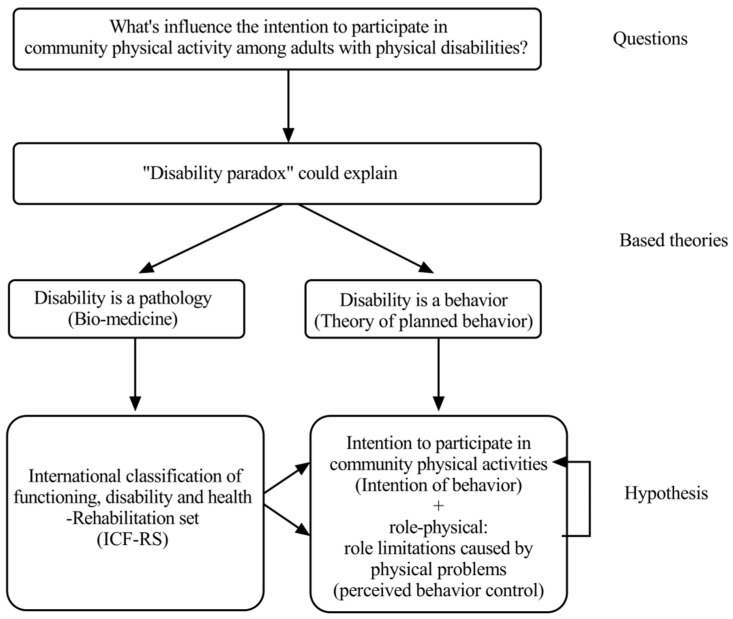
The rationale of this study.

**Figure 2 jpm-12-01832-f002:**
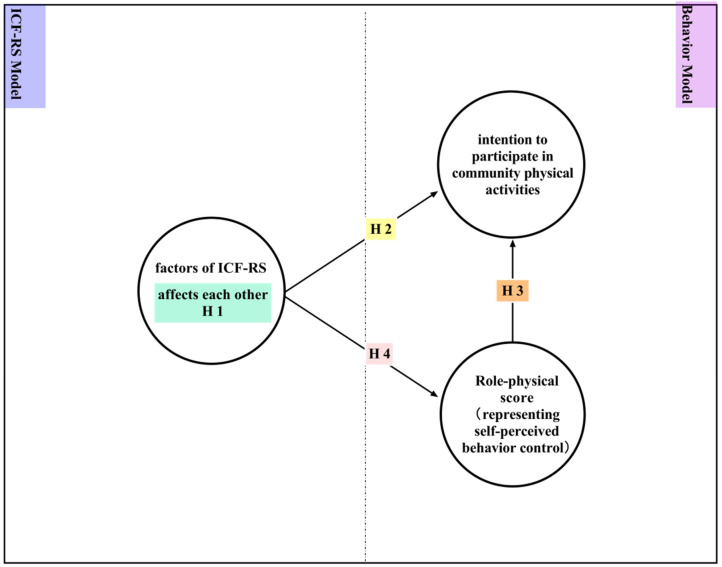
Hypothesized integrated ICF-RS and behavior model for predicting an adult with a physical disability’s intention to participate in community physical activities.

**Figure 3 jpm-12-01832-f003:**
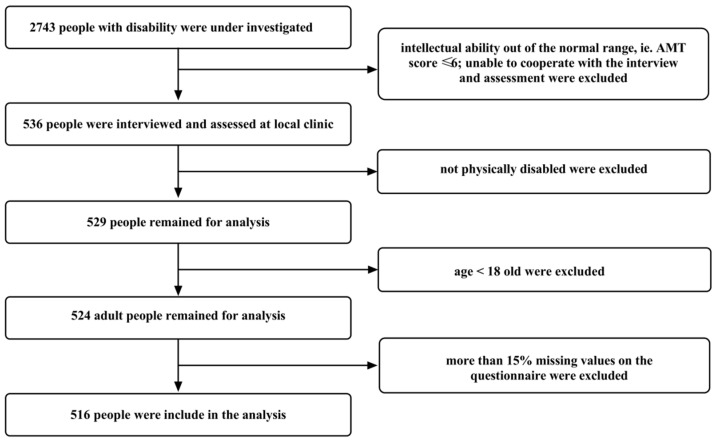
Inclusion and exclusion flowchart. Note: AMT = Abbreviated Mental Test.

**Figure 4 jpm-12-01832-f004:**
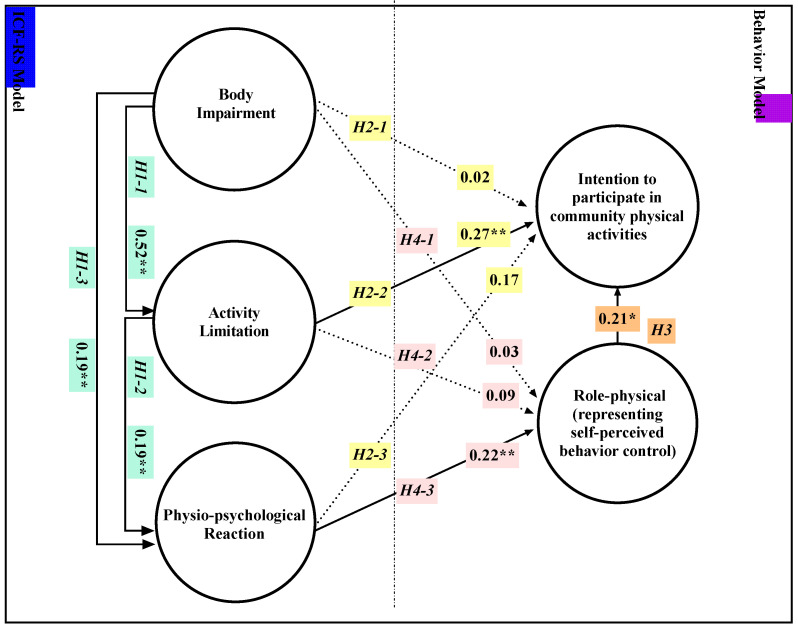
Standardized coefficients between factors for the final integrated model predicting intention to participate in community physical activities. ** Indicates a coefficient significant at the *p* ≤ 0.01 (* *p* ≤ 0.05) level of confidence. Solid lines indicate significant correlation. Dashed lines indicate no significant correlation, disallowing the hypothesis. H = hypothesis.

**Table 1 jpm-12-01832-t001:** Demographic characteristics of the participants.

Variable	N (%)	Missing Records (N)
Sex, Male	368 (71.5%)	1
Age, years		16 *
≤39 years	81 (16.2%)	
40–59 years	241 (48.2%)	
≥60 years	178 (35.3%)	
Education		6
None	49 (9.6%)	
Elementary	196 (38.4%)	
Secondary or over	265 (52.0%)	
Marital status		10
Married	371 (73.3%)	
Unmarried	135 (26.7%)	

Note: * Cases with missing age were not excluded if they could be considered adults based on their education or marital status.

**Table 2 jpm-12-01832-t002:** Standardized factor loading of the final model’s variables.

Latent Factors	Items/Variants	Standardized Factor Loading
Body Impairment		
	b730 Muscle power functions	0.846
	b710 Mobility of joint functions	0.837
Activity Limitation		
	d420 Transferring oneself	0.772
	d450 Walking	0.674
	d520 Caring for body parts	0.855
	d530 Toileting	0.891
	d540 Dressing	0.776
	d550 Eating	0.692
Physio-psychological Reaction		
	b134 Sleep functions	0.711
	b152 Emotional functions	0.679
	b280 Sensation of pain	0.533
Role-physical (representing self-perceived behavior control)		
	Perceived limited by time taken in tasks	0.872
	Perceived to have accomplished less	0.874
	Perceived limit on types of tasks	0.858
	Perceived difficulty with tasks	0.827
Intention to participate in community physical activities		
	whether willing to participate	0.507
	whether have a chance to participate	0.454

## Data Availability

The datasets used and analyzed are available from the corresponding author (T.Y.) on request.

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
