# Peer review of "Predicting Intention to Participate in Community Physical Activities for Adults with Physical Disabilities"

_jpm, 2022, doi:10.3390/jpm12111832_

Round 1
Reviewer 1 Report
Dear authors,
Thank you for the opportunity to review your paper.
The manuscript is well written and follow a rigorous mythological design. However, a few issues should be considered in the revised version:
· The rationale behind each hypothesis should be strengthened with theoretical arguments and results of previous research results.
· It is said that the survey was conducted between May and December 2019. Please explain if and how data collection was affected by the COVID 19 pandemic.
· Data in Table 2 may be better arranged as to not occupy as much space. Moreover, it may be moved in Appendices.
· Line 174: I suppose those words don’t belong there.
· In the discussions section, I suggest that the results would be discussed against the theoretical approaches and results of similar research.
· The authors should better outline the implications of the results.
Reviewer 2 Report
Journal: JPM (ISSN 2075-4426)
Title: Predicting adult with physical disability's intention to participate in community physical activities
Reviewer Comments:
Dear Editor and authors, thank you for the opportunity to review this manuscript.
The study was written clearly and good logical structure. It has demonstrated merits in terms of its scientific contributions. The authors integrated the ICF-RS items and theory of planned behaviour to determine the intention to participate in community activities. Methodology: No issues. The discussion was comprehensive and well-written with limitations and reference to previous studies.
Regarding their revised model, may I suggest the authors to consider adding one short section on how their model could be used to advise further research or clinical application? This will certainly benefits the clinicians in this field.
I have also provided other minor suggestions to improve the manuscript:
· L174: There seems to be a typo or misalignment here. Please amend it.
· Table 3: Please consider realigning the items to the latent factors so that it will be visually easier for readers to understand. There is no issue with the content, just the presentation.
· L303: Can you rephrase the subtitle, 4.4 as I do not really understand what this means?
· L324: There is a typo error here: “explanitory”.
Reviewer 3 Report
The study you present seems interesting to me.
I would like you to add a paragraph in the last part of the discussion that highlights the importance of what you propose in practice (highlight the importance of your study).
Everything else became clear to me. Congratulations.
Round 2
Reviewer 1 Report
I appreciate the improvements of the paper